# A-FABP in Metabolic Diseases and the Therapeutic Implications: An Update

**DOI:** 10.3390/ijms22179386

**Published:** 2021-08-30

**Authors:** Hang-Long Li, Xiaoping Wu, Aimin Xu, Ruby Lai-Chong Hoo

**Affiliations:** 1Division of Clinical Pharmacology and Therapeutics, Department of Medicine, LKS Faculty of Medicine, The University of Hong Kong, Hong Kong 999077, China; ronli22@hku.hk (H.-L.L.); amxu@hku.hk (A.X.); 2Department of Pharmacology and Pharmacy, LKS Faculty of Medicine, The University of Hong Kong, Hong Kong 999077, China; xpwu@connect.hku.hk; 3State Key Laboratory of Pharmaceutical Biotechnology, LKS Faculty of Medicine, The University of Hong Kong, Hong Kong 999077, China

**Keywords:** A-FABP, metabolic syndrome, cardiovascular diseases

## Abstract

Adipocyte fatty acid-binding protein (A-FABP), which is also known as ap2 or FABP4, is a fatty acid chaperone that has been further defined as a fat-derived hormone. It regulates lipid homeostasis and is a key mediator of inflammation. Circulating levels of A-FABP are closely associated with metabolic syndrome and cardiometabolic diseases with imminent diagnostic and prognostic significance. Numerous animal studies have elucidated the potential underlying mechanisms involving A-FABP in these diseases. Recent studies demonstrated its physiological role in the regulation of adaptive thermogenesis and its pathological roles in ischemic stroke and liver fibrosis. Due to its implication in various diseases, A-FABP has become a promising target for the development of small molecule inhibitors and neutralizing antibodies for disease treatment. This review summarizes the clinical and animal findings of A-FABP in the pathogenesis of cardio-metabolic diseases in recent years. The underlying mechanism and its therapeutic implications are also highlighted.

## 1. Introduction

Fatty acid-binding proteins (FABPs) belong to the lipocalin family and are a group of 14–15 kDa proteins involved in intracellular lipid homeostasis as well as metabolic and inflammatory pathways [1,2]. They are ubiquitously expressed in body organs/tissues, including the liver, intestine, heart, adipose tissue, epidermis, ileum, brain, myelin, and testis [2]. The isoforms were named according to the expression predominance or the organ where they were first identified. Different isoforms exhibit unique expression patterns and are mostly abundant in tissues involved in lipid metabolism [1,2]. Despite the considerable variation in protein sequences, FABPs share similar tertiary structures: a beta-barrel domain and an internal water-filled cavity, where hydrophobic ligands, such as long-chain fatty acid and eicosanoids, are bound to with high affinity [3,4]. FABPs function as lipid chaperones to transport lipids to specific organelles, such as the endoplasmic reticulum (ER) and mitochondria, within the cell [1].

Adipocyte-FABP (A-FABP, also referred to aP2 or FABP4) is one of the most abundant cytosolic proteins in adipocytes and is also expressed in macrophages and endothelial cells [5]. Expression of A-FABP is induced during adipocyte differentiation [6]. Functionally, A-FABP regulates lipolysis by activating hormone-sensitive lipase (HSL) and by transporting HSL-derived fatty acids in adipocytes [7,8]. A-FABP is also elevated during the differentiation of monocytes into macrophages [9]. In macrophages, its expression can be induced by fatty acids, lipopolysaccharide (LPS), toll-like receptors (TLRs) agonist [10], oxidized low-density lipoprotein (oxLDL) [11], and advanced glycation end products (AGEs) [12]. A-FABP also promotes cholesterol ester accumulation and foam cell formation [13,14]. In endothelial cells, increased A-FABP induces endothelial dysfunction by modulating endothelial *NO* synthase (eNOS) nitric oxide (NO) signaling [15]. In addition to its intracellular effects, A-FABP is also recognized as a hormone that is released into the circulation and exerts effects on target tissues, such as adipose tissue and endothelium [16]. 

A-FABP contributes to the pathogenesis of a great variety of disorders/diseases, including metabolic syndrome [17,18,19], atherosclerotic diseases [20], heart failure [21], and non-alcoholic steatohepatitis [22]. Recent findings also demonstrated the pathological roles of A-FABP in ischemic stroke and liver fibrosis [23,24] and implicated the potential of A-FABP as a sensitive predictor of the outcome of alcohol-induced acute-on-chronic liver failure [25]. In addition, the involvement of A-FABP in various cancer types, including bladder cancer [26,27,28,29], prostate cancer [30,31], breast cancer [32], ovarian cancer [33,34], cholangiocarcinoma [35], hepatocellular carcinoma [36,37,38], and leukemia [39,40] was also reported. 

The potential of targeting A-FABP as a therapeutic strategy was elucidated in animal studies [41,42,43] (Appendix A), while its clinical therapeutic implications are unclear [1,44]. This review aims to evaluate and summarize the existing evidence of A-FABP in the metabolic syndrome and cardiovascular diseases, and its potential therapeutic implications. 

## 2. Metabolic Syndrome

Metabolic syndrome refers to a cluster of cardiovascular risk factors, including central obesity, insulin resistance, dyslipidemia, and hypertension [45]. A-FABP is a predictive circulating biomarker of components of metabolic syndrome [17,18]. Over the past 5 years, novel discoveries regarding the role of A-FABP in metabolic syndrome have been made (Table 1).

### 2.1. Central Obesity

Central obesity, characterized by excessive accumulation of fat, especially in the abdominal region, resulting from a prolonged positive energy imbalance [62], is the major risk factor of cardiovascular diseases (CVD) [62]. Obesity is a chronic low-grade inflammatory state in which dysfunctional adipose tissue disrupts metabolic homeostasis through altering the secretion of cytokines and hormones, such as A-FABP, and impairing the metabolism of non-esterified fatty acid (NEFA) [62,63]. 

Xu et al. were among the first to show that circulation A-FABP levels were significantly higher in obese than in lean subjects [17,18]. A-FABP was positively correlated with indicators of adiposity, including body mass index (BMI), fat percentage, waist circumference, and waist-to-hip ratio [17,18,64,65,66,67]. A recent study identified that the higher serum A-FABP was negatively associated with the sensitivity to the thyroid hormone, which suggested that A-FABP may mediate the “cross-talk” between adipose tissue and the thyroid system [68]. In obese patients with concomitant type 2 diabetes, A-FABP levels were further elevated and significantly correlated with levels of inflammatory cytokines, including C-reactive protein and interleukin-6 (IL-6) [46]. In a cohort of obese women, a positive correlation between A-FABP and tumor necrosis factor (TNF) receptors was identified [69]. This evidence indicates the crucial role of A-FABP in mediating a pro-inflammatory state in obesity.

In addition to the clinical studies, animal and in vitro studies further revealed the sophisticated role of A-FABP in adipose tissue. In diet-induced or genetic models of obesity, A-FABP-deficient mice exhibited higher adiposity while they were more “metabolically healthy” with improved glucose and lipid metabolism [70,71] when compared to their wild-type littermates. Knockdown of A-FABP using the RNA interference technique also enhanced high-fat diet–induced body weight in mice, but the improvement in glucose metabolism was not observed possibly due to a partial knockdown of A-FABP using RNA interference compared to A-FABP knockout [72]. The increased susceptibility to diet-induced obesity despite protection against the development of insulin resistance might be attributed to the reduced lipolysis efficiency and improved insulin secretory response in A-FABP-deficient mice [8,73,74]. Without altering the fatty influx/esterification and the expression of lipolysis-related proteins, A-FABP-deficient adipocytes exhibited diminished lipolysis and attenuated FFA efflux under basal condition or stimulation of isoproterenol or dibutyryl-cAMP [73,74]. Mechanistically, by interacting with hormone-sensitive lipase (HSL), the critical enzyme of lipolysis, A-FABP facilitates the out-trafficking of HSL-derived fatty acids, thus preventing the feedback inhibition of HSL by high FFA levels within cells [8]. This mechanism implicated the beneficial physiological effect of A-FABP in the context of starvation, which facilities lipid utilization. However, under obese conditions, the abundant A-FAPB provokes lipolysis leading to ectopic accumulation of lipids in other organs [75]. A-FABP-deficient mice also exhibited impaired insulin secretion in response to beta-adrenergic stimulation [74], whereas A-FABP was demonstrated to regulate the FFA metabolism by altering the composition (reduction in both stearic and cis-11-eicoseneic acids and an increase in palmitoleic acid), thereby modulating the adipo-pancreatic axis-mediated insulin secretion [74]. 

In addition to the studies using A-FABP-deficient mice/adipocytes, treatment recombinant of A-FABP protein also demonstrates its role in regulating lipolysis and inflammatory response. During differentiation of 3T3L1 preadipocytes to adipocytes, exogenous treatment of A-FABP inhibited the accumulation of lipids and suppressed the transcription of adipogenic marker genes, including PPARγ, C/EBPα, adiponectin, and A-FABP itself. On the other hand, the enhanced lipolysis was evidenced by the elevated levels of adipose triglyceride lipase (ATGL) and phosphorylated HSL (pHSL), the hallmark of lipolysis regulation [51]. Moreover, upregulation of MCP-1, TNFα, and IL-6 in adipocytes upon A-FABP treatment further implicates the role of A-FABP in pro-inflammatory response [51]. Mechanistically, A-FABP regulates lipolysis and inflammatory response through activation of p38/HSL and p38/NF-kB signaling pathways, respectively [51]. This study for the first time implicated the direct negative feedback regulation of adipocyte-derived A-FABP on adipose tissue expansion as well as its role in the initiation of adipose tissue inflammation. 

Numerous studies have demonstrated the beneficial effect of enhanced adaptive thermogenesis in improving whole-body metabolism and body weight loss [76]. A-FABP was shown to stimulate adaptive thermogenesis [49]. On one hand, circulating A-FABP transports FFA to brown adipocytes as an energy substrate for FFA oxidation. On the other hand, without altering the activity of sympathetic nervous system, A-FABP regulates the conversion of inactive thyroxine (T4) to active triiodothyronine (T3) in brown adipose tissue (BAT) by inducing the expression of enzyme type II iodothyronine deiodinase through promoting the proteasomal degradation of LXRα [49], which is a crucial step in the activation of BAT-mediated thermogenesis [77]. Furthermore, replenishment with recombinant A-FABP increased whole-body energy expenditure by 1.5-folds in A-FABP-deficient mice when compared to those treated with vehicles [49]. This study emphasized the physiology role of circulating A-FABP in BAT-mediated adaptive thermogenesis implicating that systematic inhibition of A-FABP in treating obesity-related disorders might cause adverse effects and the tissue-specific function of A-FABP warrants further investigation. 

### 2.2. Insulin Resistance

Insulin resistance refers to the impaired response of targeted cells to insulin action. Among the risk factors, obesity is the most critical one, as the aberrant release of adipose tissue-derived NEFA, glycerol, adipokines, and proinflammatory cytokines contribute to insulin resistance and eventually causes pancreatic β-cell dysfunction [78].

Clinical studies identified strong positive correlations between A-FABP, insulin resistance, and type 2 diabetes [17,18,22,65,66,67,79,80,81,82], suggesting A-FABP as a biomarker of insulin resistance. Carriers of genetic variant (rs77878271) of T-87C polymorphism in the functional promoter of A-FABP gene with reduced A-FABP expression had a lower risk of developing type 2 diabetes [83]. A study investigating the effects of exercise training on insulin resistance in middle age obese men also showed that improvement in glucose metabolism was significantly correlated with reduction in circulating A-FABP [47]. 

Animal studies also supported that A-FABP acts as a mediator of insulin resistance. In the context of DIO, A-FABP-deficient mice showed improvement in insulin sensitivity and glucose-stimulated insulin secretion [70,84,85]. The protection against the development of insulin resistance by A-FABP deficiency could be attributed to several mechanisms. A-FABP deficiency generates a lipid environment that is highly favorable for insulin action: mice lacking A-FABP had altered lipid composition in muscular tissue (upregulation of shorter chain [12:0 and 14:0] fatty acids, and downregulation of longer chain [16:0 and 18:0] fatty acids), leading to an upregulation in insulin-stimulated phosphorylation of Akt and protecting against high-fat-induced insulin resistance, thus enhancing the insulin signaling cascade [84]. The basal level and leptin-stimulated activity of AMP-K-α1, an important energy sensor in muscular tissue, were also elevated in A-FABP-deficient mice when compared to wild-type mice [84]. Furthermore, A-FABP-deficient DIO mice not only exhibited attenuation in beta-adrenergic-stimulated lipolysis [74,85] but also shown reduced secretion of inflammatory cytokines, such as TNFα [70], when compared to their relative controls. The treatment of human THP-1 macrophages with intermittent high glucose stimulated the expression of A-FABP, which subsequently mediated inflammatory cytokine (TNF-α and IL-1β) secretion through activating TLR4/p-JNK signaling cascade [50], which implicates the additional regulatory effect of A-FABP on inflammation in response to glucose fluctuation under insulin resistance.

### 2.3. Dyslipidaemia

Dyslipidemia includes increased low-density lipoproteins (LDLs), decreased high-density lipoproteins (HDLs), and increased fasting and postprandial triglyceride (TG)-rich lipoproteins (very-low-density lipoproteins [VLDL] and chylomicrons) [86,87]. The mechanism through which dyslipidemia develops is closely related to insulin resistance [88], as unrestrained lipolysis leads to an increased hepatic flux of FFA, contributing to increased hepatic TG, hence VLDL production [89,90]. Lipid overload in non-adipose tissues causes cellular dysfunction and apoptotic cell death, leading to lipotoxicity [91]. Clinical studies have shown a positive correlation of A-FABP with hypertriglyceridemia and LDL, as well as an inverse correlation with HDL [17,18,65,66,67,79,81,83].

Aside from the effects of A-FABP on lipolysis, it also potentiates dyslipidemia-related lipotoxicity and chronic inflammation. In macrophages, A-FABP promotes toxic lipid-induced ER stress, leading to the exaggeration of inflammation via suppressing Janus kinase (JAK) 2-dependent autophagy [48]. Palmitic acid-mediated elevation of A-FABP in macrophages downregulated the autophagy-related protein 7, leading to the suppression of autophagy and increase of ER stress [48]. On the contrary, in A-FABP deficient macrophages, the phagocytic activity was significantly higher, the LPS-INFγ-induced M1 macrophage polarization was attenuated, while the IL4-induced M2 markers were markedly enhanced [48]. These findings implicate that A-FABP is a critical player in lipotoxicity-related inflammatory disorders.

### 2.4. Hypertension

Hypertension is defined as a prolonged period of high blood pressure with systolic blood pressure >140 mmHg or diastolic blood pressure >90 mmHg [92]. Multiple mechanisms, including endothelial dysfunction, endothelin-1 overexpression due to dysregulated secretion of adipokines [93,94], and the associated unrestricted vasoconstriction [95], contribute to the development of hypertension. Other contributory mechanisms include the anti-natriuretic effect of insulin [96,97], a positive feedback loop between an upregulated renin–angiotensin system and an overactive sympathetic nervous system [98,99] as well as a dysregulated autonomic nervous system–associated volume overload. A-FABP was identified to be a key mediator in these pathophysiological pathways involved in the development of hypertension, including being associated with endothelial dysfunction through mediating endothelial nitric oxide pathways [100] as well as activation of the sympathetic nervous system [101]. Indeed, ample human evidence has demonstrated the correlation between A-FABP and blood pressure. Circulating levels of A-FABP are positively correlated with systolic/diastolic blood pressure [17,18,67,79]. Ota et al. further identified that increased circulating A-FABP was associated with increased blood pressure, and the elevation of A-FABP was predisposed by a family history of hypertension [81].

### 2.5. Clinical Diagnostic and Prognostic Implications

A series of human studies have provided concrete evidence for the association between levels of A-FABP and metabolic syndrome in a wide variety of populations, including both Asians and Caucasians. A-FABP is an independent biomarker of metabolic syndrome and may be useful in diagnosing and predicting the risk of developing metabolic syndrome [17,64]. A 5-year prospective study showed that levels of A-FABP predicted the development of metabolic syndrome independent of insulin resistance and adiposity [18]. A 10-year prospective study found that serum A-FABP levels could predict the development of type 2 diabetes [19]. These large-scale long-term human studies support the utility of A-FABP as a diagnostic and prognostic biomarker.

## 3. Cardiovascular Diseases

Cardiovascular disease (CVD), including myocardial infarction (MI), stroke, and peripheral vascular disease, are the leading causes of death worldwide [102]. Atherosclerosis is the predominant cause of these medical conditions [103]. In multiple long-term follow-up studies of various patient cohorts, circulating levels of A-FABP were shown to predict the development of CVD and cardiovascular mortality [52,79,104,105]. The important findings on the role of A-FABP in CVD in recent years are summarized in Table 1.

### 3.1. Atherosclerosis

Atherosclerosis is characterized by the narrowing/hardening of arteries caused by the buildup of plaque, which is made up of substances, including fats and cholesterol [103]. Risk factors, such as endothelial dysfunction, dyslipidemia, hypertension, and type 2 diabetes, contribute to the pathogenesis of atherosclerosis [106,107]. The mechanisms through which atherosclerosis develops are manifold. Mechanistically, endothelial dysfunction-associated reduction of nitric oxide (NO) production, generation of reactive oxidative species (ROS), and increase in oxLDL trigger an inflammatory response, thus leading to atherosclerosis [15]. Apart from metabolic risk factors, smoking is also another major contributory factor for the development of atherosclerosis through multiple mechanisms, including promoting the formation of ROS, causing endothelial dysfunction, and inducing a systemic inflammatory state [108]. Indeed, A-FABP has been implicated in the development of endothelial dysfunction through mediating production of nitric oxide as well as systemic inflammation [100]. Although no prior studies have comprehensively evaluated the association between A-FABP and cigarette smoking, A-FABP levels were found to be higher among women exposed to polycyclic aromatic hydrocarbons, a major constituent in cigarettes that triggers an inflammatory reaction [109].

Human studies have shown a strong association between A-FABP and atherosclerotic conditions (Table 2). Circulating levels of A-FABP are closely associated with carotid intima-media thickness (CIMT), a well-established marker of atherosclerosis [57,110,111]. In Chinese cohorts, serum A-FABP levels were independently associated with CIMT in women but not in men [110,111], which might be due to lower levels of A-FABP in men. Higher basal levels of A-FABP were also associated with larger changes in CIMT, suggesting that A-FABP predicts the progression of atherosclerosis [57]. Moreover, expression of A-FABP was elevated in carotid plaques in patients with CVD and was associated with plaque vulnerability [112,113]. Patients with higher baseline levels of A-FABP had an increased risk of subclinical atherosclerosis in a cohort of Chinese patients [114]. A recent study also showed a significant association between A-FABP expression levels in epicardial adipose tissue and the extent of coronary atherosclerosis in patients with metabolic syndrome and coronary artery disease (CAD) [60]. On the contrary, patients with T-83C polymorphism exhibited lower A-FABP expression in carotid plaques and adipose tissue, had a lower prevalence of carotid plaques, reduced CIMT, and a reduced risk of developing CAD and MI [83,113]. By using coronary thrombectomy specimens from patients with acute myocardial and autopsy coronary artery and specimens from patients with ischemic heart disease, the elevated expression of A-FABP was identified in macrophages within atherosclerotic lesions and epicardial/perivascular adipocytes [53]. 

Animal studies showed that A-FABP mediates the pathogenesis of atherosclerosis via inducing endothelial dysfunction [15,100], vascular smooth muscle cell invasion [53,56,115], foam cell formation [11,100], and inflammatory response [13,115]. On the contrary, A-FABP deficiency in apolipoprotein E (ApoE)–deficient mice protected against atherosclerosis [9,116,117] and even high-fat diet–induced advanced atherosclerosis [118].

In ApoE^-/-^ mice, who developed atherosclerotic plaques spontaneously, the presence of A-FABP was observed in the aortic endothelium from 12 weeks, while pharmacological inhibition of A-FABP by BMS309403 significantly improved endothelial function through rescuing the eNOS-NO signaling pathway [15]. Consistent with in vivo studies, lipid-induced elevation of A-FABP in human microvascular endothelial cells was accompanied with reduced phosphorylated eNOS and NO production, which was reversed upon BMS309403 treatment [15]. A-FABP was also induced in endothelial cells of the hyperplastic neointima of mice subjected to wire-induced vascular injury [56]. In human coronary artery endothelial cells (HCAECs), A-FABP expression was induced upon vascular endothelial growth factor (VEGF) or hydrogen peroxide (H_2_O_2)_ treatment [56]. Adenovirus-mediated A-FABP overexpression inhibited the VEGF or insulin-stimulated eNOS phosphorylation and induced pro-inflammatory cytokine/adhesion molecules expression [56]. In human vascular endothelial cells (HUVECs), exogenous treatment of recombinant A-FABP (rA-FABP) also reduced the level of phosphorylated eNOS [53]. Upon palmitic acid treatment, rA-FABP not only further reduced the p-eNOS, but also upregulated the expression of pro-inflammatory cytokines, including MCP-1, IL-6, and TNFα [53]. Moreover, r-A-FABP treatment impaired the insulin-mediated eNOS pathway in vascular endothelial cells by inhibiting insulin receptor substrate 1 (IRS1) and Akt activation [100], which implicates the mechanistic linkage between circulating A-FABP and endothelial cell dysfunction in diabetes. Furthermore, rA-FABP stimulated cell proliferation and migration of human coronary artery smooth muscle cells (HCASMCs) through upregulating cell cycle regulations (cyclin D1, CCL2, and MMP2) via activating c-jun and c-myc in a MAPK-dependent manner [116]. It also induced pro-inflammatory cytokine expression in HCASMCs [53,56,115]. 

A-FABP plays a critical role in foam cell formation and the subsequent development of cholesterol-rich lesions. In ApoE^-/-^ mice with macrophage-specific A-FABP deficiency, the reduction in atherosclerotic lesions was comparable with ApoE^-/-^ mice with global A-FABP deficiency, suggesting the independent role of macrophage A-FABP in the pathogenesis of atherosclerosis [9]. In macrophages, A-FABP is induced by LPS [9,13,14] through activating JNK-c-Jun signaling [13] and oxLDL via activating NF-κB and PKC signaling pathways and PPARγ [11,117]. A-FABP-deficient macrophages not only exhibited a reduced capacity for inflammatory cytokine production [9,14] but also showed reduced total cholesterol and cholesterol ester content, due to accelerated cholesterol efflux [14]. A-FABP regulates cholesterol trafficking through mediating the PPARγ-LXRα-ABCA1 pathway [14]. 

In addition, A-FABP promotes atherosclerosis by mediating inflammatory responses in macrophages, T cells, and dendritic cells [13,115]. A-FABP-deficient mice exhibited a significantly reduced expression of inflammatory cytokines [119] and inflammasome activation [54,120]. In macrophages, A-FABP mediates the inflammatory response induced by various stimulators. In response to LPS, A-FABP forms a positive feedback loop with JNK/AP-1, thereby upregulating the expression of inflammatory cytokines in macrophages [13]. On the other hand, in response to LPS or CD154 stimulation, A-FABP activates the IκB/NF-κB pathway, thus inducing the inflammatory activity of macrophages [14]. Upon the stimulation of palmitic acid, elevated A-FABP provoked the toxic lipid-induced ER stress via inhibiting the JAK2-dependent autophagy, which in turn triggered M1 macrophage polarization and the inflammatory cytokine expression [48]. In response to intermittent high glucose treatment, the activation of TLR4/p-JNK cascade upregulated both A-FABP expression and inflammatory cytokine secretion, which implicates that A-FABP might also be involved in the glucose fluctuation-associated inflammatory response [50]. A-FABP also activates the IκB/NF-κB pathway in T cells and dendritic cells, thus inducing inflammatory cytokine secretion [115].

### 3.2. Ischemic Stroke

A-FABP has long been implicated in the progression and severity of ischemic stroke (IS). There is concrete evidence showing the association between A-FABP and the risk factors/poor prognostic markers of ischemic stroke, such as type 2 diabetes, hypertension, dyslipidemia, arterial stiffness, higher levels of high-sensitivity C-reactive protein, and cerebral embolization from carotid atherosclerosis [17,121,122,123]. Circulating A-FABP levels are elevated after acute ischemic stroke and are positively correlated with early death and poor functional outcome [58,124]. 

A recent finding demonstrated that A-FABP mediates the ischemia-induced blood–brain barrier (BBB) disruption, contributing to cerebral ischemia injury [23]. Briefly, by using middle cerebral artery occlusion (MCAO)–induced ischemic stroke model, researchers identified the elevation of A-FABP in peripheral monocytes and microglia [23]. Mechanistically, on one hand, via activating the JNK/c-Jun signaling cascade, A-FABP upregulates the expression of matrix metalloproteinase-9 (MMP-9) in bone marrow-derived macrophage (BMDM). BMDM-derived MMP-9 degrades the extracellular matrix and tight junction proteins of the BBB and allows the infiltration of blood-borne immune cells into the brain [23]. Elevated A-FABP also enhances the production of inflammatory cytokines from microglia, which further provoke ischemic injury. The upregulation of cytokines, such as TNFα and IL-1β, might also induce MMP-9 expression [125]. All these factors lead to post-ischemic inflammation and poor functional recovery [23,126,127]. 

### 3.3. Heart Failure

Heart failure is a chronic syndrome representing the inability of the heart to pump sufficient blood to meet the oxygen demand of the body [128]. Coronary atherosclerosis and components of metabolic syndrome are the risk factors contributing to left ventricular systolic/diastolic dysfunction, increasing the risk of heart failure [128].

Human studies have identified an association between A-FABP and various cardiac abnormalities that predispose heart failure. Circulating A-FABP levels were associated with myocardial perfusion abnormalities [129], positively correlated with left ventricular mass and hypertrophy [126,127] and were strongly correlated with N-terminal fragment of pro-B-type natriuretic peptide (NT-proBNP), a well-established biomarker of heart failure [130]. Circulating A-FABP was identified as an independent risk factor in left ventricular dysfunction [21,131] and a 10-year prospective study of over 4000 patients showed that A-FABP levels predicted the development of heart failure [132]. A recent study identified that A-FABP levels predicted the development of left ventricular hypertrophy, diastolic dysfunction, and adverse cardiovascular events in patients with type 2 diabetes without established cardiovascular diseases in a median follow-up duration of 28 months [59]. Further, another recent study identified a significant association between A-FABP levels and key hemodynamic indices and showed that the accuracy of HF risk classification models could be improved by incorporating A-FABP levels [61]. Thus, A-FABP has been proposed as a biomarker and predictor for heart failure [133].

In animal studies, several mechanisms linking A-FABP and heart failure have been proposed. A-FABP exhibits cardio-depressant properties, as it suppresses myocardial contractility, contributing to systolic dysfunction [134,135]. Without modulating the action potential duration and the L-type Ca^2+^ channel activity, adipocyte-derived A-FABP represses the cardio-depressant activity in a dose-dependent manner [134]. Specifically, the N-terminal of A-FABP confers cardio-depressive properties [134]. The mechanism of cardio-depressant effects may be related to the fact that fatty acids are the main energy source for cardiomyocytes and that A-FABP binds fatty acid with high affinity, thus disrupting energy supply and contractile function [136]. The overexpression of A-FABP exacerbated pressure-induced heart hypertrophy in mice by upregulating the expression of cardiac hypertrophic marker genes, while treatment with BMS309403 reversed the condition [55]. A-FABP deficiency also attenuated ischemia/reperfusion-induced myocardial injury and improved left ventricular function in mice [137].

## 4. Therapeutic Implications

As A-FABP is involved in various diseases and represents a therapeutic target, multiple small molecule inhibitors, including BMS309403, HTS01037, and other triazolopyrimidine derivatives [1,41,42], which inhibit A-FABP by various modes of binding, and polyclonal [138] and monoclonal neutralizing antibodies such as CA33 and 2E4 [139,140], have been developed, aiming to inhibit A-FABP activity. A fluoroquinolone antibiotic (levofloxacin) and a uricosuric agent (benzbromarone) were also found to exhibit anti-A-FABP properties [141,142].

### 4.1. Small Molecule Inhibitors of A-FABP

BMS309403 is by far the most studied selective inhibitor of A-FABP. It is a synthetic small molecule that competes with endogenous FFA to bind to the fatty-acid binding pocket of A-FABP [1]. Type 2 diabetes, insulin resistance, and glucose metabolism were alleviated in both genetically obese and DIO mouse models upon treatment with BMS309403 [1]. Compared to vehicle-treated control, macrophage infiltration in adipose tissue was less prominent in BMS309403-treated mice, which was accompanied with reduced levels of proinflammatory cytokines (MCP-1, IL-6, and TNFα) in adipose tissues [143]. 

For atherosclerosis and CVD, treatment with BMS309403 significantly reduced the atherosclerotic lesion area in the proximal aorta in diabetic ApoE^-/-^ mouse models [143]. It also reversed the A-FABP-mediated dysfunction of eNOS/NO signaling pathway, stimulated endothelial relaxation, and improved endothelial function in ApoE^-/-^ mouse models after a 6-week regime [15]. In addition, treatment of BMS309403 led to a significant reduction in expression of cardiac hypertrophic genes in transgenic A-FABP mice [55] and reduced ischemia/reperfusion injury, brain oedema, and neurological deficits in mice subjected to middle cerebral artery occlusion [23]. Other potential long-term benefits of A-FABP inhibition in chronic management of ischemic stroke were also suggested by the higher survival rate among mice treated with BMS309403 [23]. However, in vitro studies indicated that application of BMS309403 might exert acute cardiac depressant effects [144]. Other off-target effects include dose- and time-dependent stimulation of glucose uptake in C2C12 myotubes via activation of the AMP-activated protein kinase (AMPK) signaling pathway, which is independent of FABPs [145]. Further investigation into the off-target effects of BMS309403 is needed.

HTS01037 is another small molecule inhibitor of A-FABP, which inhibits lipolysis in adipocytes [42]. It inhibited LPS-stimulated inflammation in macrophages and attenuated pro-inflammatory NF-κB signaling [42,146]. However, the binding affinity to A-FABP and the potency of HTS01037 are lower compared to BMS309403 [42]. Two of the other triazolopyrimidine derivatives, known as Compounds 2 and 3, were also able to occupy the ligand-binding pocket of A-FABP and exhibited high affinity with A-FABP [41]. Compared to BMS309403, they have higher potency in inhibiting lipolysis, and they led to a comparable suppression in the secretion of MCP-1 from macrophages [41].

### 4.2. Antibodies

As the suboptimal specificity of small molecule/chemicals may lead to off-target effects, there is a growing trend in the development of antibodies targeting A-FABP, which may have a higher specificity. The alleviation of insulin resistance and glucose metabolism, as well as reduction in fat mass and steatosis, were observed in obese mice treated with monoclonal antibody CA33, which acts against A-FABP [139]. The monoclonal A-FABP neutralizing antibody 2E4 suppressed the expression of pro-inflammatory cytokines in mice with dietary obesity [140]. Treatment of polyclonal antibody against A-FABP suppressed hepatic glucose production and reversed diabetes in obese mice [138]. Improved endothelial function, as well as reduced proliferation and migration of coronary smooth muscle cells after angioplasty, were observed in mouse models after treatment with polyclonal antibody of A-FABP, suggesting its potential application for the prevention of post-angioplasty vascular restenosis [56]. This evidence shows that neutralization of A-FABP activity is potentially effective and beneficial in metabolic and atherosclerotic diseases.

### 4.3. Other Drugs/Strategies

Other drugs that are approved by the Food and Drug Administration, such as levofloxacin, a broad-spectrum fluoroquinolone used to treat infectious diseases, and benzbromarone, a uricosuric agent used to treat gout, exhibit A-FABP-inhibitory properties [141,142]. For instance, benzbromarone reduced blood glucose levels in obese mouse models with type 2 diabetes [142]. Other novel strategies, such as short hairpin RNA (shRNA) and RNA interference (RNAi) have also been developed to target A-FABP. shRNA-mediated A-FABP knockdown improved insulin sensitivity and glucose tolerance in mouse models [147]. RNAi significantly reduced the levels of A-FABP in adipocytes of diet-induced mice by 70–80% [72]. Although none of the above studies of A-FABP inhibition was carried out in preclinical trials, these results suggested the potential of A-FABP inhibition as a therapeutic strategy in a multitude of diseases.

### 4.4. Major Concerns

Although inhibition of A-FABP appeared to be beneficial in mouse disease models, BMS309403 has been shown to exert off-target effects [144,145], and its safety in humans is yet to be determined. BMS309403 is not classifiable according to the Globally Harmonized System of Classification and Labelling of Chemicals (GHS), based on the currently available data and evidence [148].

Levels of A-FABP can be affected by drugs that are commonly used in clinical practice. Lipid-lowering agents, including atorvastatin and omega-3 fatty acid ethyl esters, reduce circulating A-FABP levels [149,150]. Angiotensin II receptor block (ARB) also reduced A-FABP levels in hypertensive subjects [151,152]. Sitagliptin, a dipeptidyl peptidase 4 inhibitor (DPP4i), is an anti-diabetic agent that also reduces levels of A-FABP [153]. On the contrary, A-FABP levels were elevated in patients treated with other anti-diabetic agents, such as canagliflozin, a sodium-glucose cotransporter 2 inhibitor, and pioglitazone, a thiazolidinedione [154,155]. In a cohort of patients with type 2 diabetes, treatment with canagliflozin paradoxically increased serum A-FABP, possibly through inducing catecholamine-mediated lipolysis [155]. These paradoxical observations again suggested that the expression and actions of A-FABP are complex when other commonly used drugs are administered. Considering the physiological role of A-FABP in adaptive thermogenesis and its related potential in protection against excessive weight gain, more research on the mechanics and dynamics of A-FABP is needed, and tissue-specific inhibition of A-FABP may be one of the key future research directions. In general, before clinical trials, further investigations into the pharmacokinetics, pharmacodynamics, and pharmacological interactions of A-FABP with other drugs are warranted.

### 4.5. Other Potential Applications of A-FABP Inhibition

PCSK9 inhibitors have been shown to be effective in improving the lipid profile [156]. PCSK9 is an enzyme that mediates the hepatic expression of LDL receptors, thereby modulating the metabolism of lipoproteins. Interestingly, levels of A-FABP were independently associated with PCSK9 levels, suggesting that A-FABP may mediate the effects of PCSK9 on LDL homeostasis [157]. These findings implicate the possibilities of PCSK9 inhibitors acting as A-FABP inhibitors and vice versa. Furthermore, PCSK9 inhibitors have been shown to reduce cardiovascular mortality [158]. The therapeutic potential of concomitant inhibition of A-FABP and PCSK9 could be further explored in future studies.

## 5. Conclusions

The associations between A-FABP, metabolic syndrome, and CVD are well established, thanks to the large number of animal and clinical studies that together dissected the expression and functions of A-FABP in these conditions (please refer to the summary figure). A-FABP was suggested as a biomarker for metabolic syndrome and CVD. Pre-clinical studies of A-FABP inhibition showed encouraging results, but there are also paradoxical results, suggesting that our understanding of A-FABP is still insufficient. In particular, issues related to the differential or even beneficial effects of A-FABP, as well as the safety of A-FABP inhibition in humans have not been well studied. Further research on the profiling of A-FABP expression and inhibition in both animal models and humans are warranted to justify A-FABP as a reliable biomarker and therapeutic target for metabolic diseases.

## Figures and Tables

**Table 1 ijms-22-09386-t001:** Summary of important novel research findings in 2016–2021.

Year	Diseases/Conditions	Subjects/Animals/Methods	Main Novel Findings	Reference
**Metabolic syndrome**				
2016	Type-2 diabetes/Obesity	48 non-obese subjects newly diagnosed with type 2 diabetes; 42 obese subjects newly diagnosed with type 2 diabetes; 30 simple obese subjects; and 30 matched normal subjects	1. Serum A-FABP levels were significantly correlated with HbA1c 2. Serum A-FABP levels correlated with levels of inflammatory cytokines (C-reactive protein and IL-6) in obese diabetic subjects	Niu G et al. [46]
2017	Obesity	22 obese middle-aged men randomized to exercise training group or control group	Exercise training reduced A-FABP concentrations and improved glucose metabolism in obese middle-aged men	Bahrami Abdehgah E et al. [47]
2017	Lipotoxicity/ ER stress/Autophagy	Macrophages isolated from A-FABP knockout mice treated with palmitic acid and/or infected with adenoviruses over-expressing A-FABP	1. Prolonged treatment of palmitic acid enhanced the expression of A-FABP associating with increased endoplasmic reticulum stress and reduced autophagic flux in macrophages 2. A-FABP suppressed PA-induced JAK-dependent autophagy thus promoted ER stress and inflammation in macrophages.	Hoo RL et al. [48]
2017	Adaptive thermogenesis	A-FABP knockout mice were infused with recombinant A-FABP after HFD for 4 weeks	1. A-FABP levels were increased in both white and brown adipose tissue in response to thermogenic stimuli 2. A-FABP deficiency impaired adaptive thermogenesis in mice, which were reversed by replenishment of recombinant A-FABP 3. A-FABP induced the expression of type-II iodothyronine deiodinase in brown adipose tissue, promoting the conversion of thyroid hormones from its inactive form T4 to active form T3, thus enhancing thermogenic activity.	Shu L et al. [49]
2018	Glucose fluctuation on macrophage inflammation	Human monocytic THP-1 cells were exposed to normal, constant high, or intermittent high glucose	1. Intermittent high glucose induced A-FABP expression and release of pro-inflammatory cytokines. Treatment with constant high glucose showed similar effects but with less evident changes. 2. Inhibition of JNK signalling pathway inhibited glucose-induced A-FABP expression and production of pro-inflammatory cytokines	Li H et al. [50]
2020	Lipolysis/ Pro-inflammation	Adipocytes were co-treated with recombinant A-FABP and A-FABP inhibitor (SB203580/I-9) or vehicle; Male mice were subcutaneous injected with recombinant A-FABP	1. Exogenous treatment of A-FABP resulted in anti-adipogenesis by inducing lipolysis (via p38/HSL signalling) and inflammation (via NF-κB signalling) 2. The pro-inflammatory and pro-lipolytic effects of exogenous A-FABP were reversed by A-FABP inhibitor	Dou HX et al. [51]
**CVD**				
2016	Cardiovascular mortality	950 male subjects with type 2 diabetes with an average follow-up for 22 years	Higher levels of A-FABP were significantly associated with higher CVD mortality	Liu G et al. [52]
2016	Coronary atherosclerosis	Human macrophages and coronary artery-derived smooth muscle cells and endothelial cells were treated with exogenous A-FABP	1. Exogenous treatment with A-FABP stimulated the inflammatory response in vascular endothelial cells in a dose-dependent manner 2. Serum A-FABP levels were correlated with coronary sinus A-FABP 3. A-FABP in coronary sinus and aortic root independently predicted severity of coronary stenosis	Furuhashi M et al. [53]
2016	Macrophage inflammation	Macrophages from A-FABP knockout or wild-type mice	1. Sirtuin 3 was upregulated in A-FABP deficient macrophages 2. Elevated sirtuin 3 attenuated lipopolysaccharide-induced expression of inflammatory cytokines, inducible nitric oxide synthase, and cyclo-oxygenase 2	Xu H et al. [54]
2016	Heart failure	Cardiomyocyte-specific A-FABP transgenic mice treated with A-FABP inhibitor (BMS309403)	1. Over-expression of A-FABP in cardiomyocytes activated ERK signaling pathway and upregulated the expression of cardiac hypertrophic marker genes 2. Aggravation of cardiac hypertrophy was alleviated with A-FABP inhibitor	Zhang J et al. [55]
2017	Vascular Injury/ Neointima formation	1. A-FABP deficient mice and relative wild-type mice subjected to wire-induced vascular injury 2. Human coronary artery endothelial cells (HCAECs) and human coronary smooth muscle cells were infected with adenovirus-overexpressing A-FABP or treated with anti-A-FABP antibody.	1. A-FABP deficient mice exhibited decreased neointima formation in response to wire-induced vascular injury. 2. Human coronary artery endothelial cells secreted A-FABP 3. Adenovirus-mediated overexpression of A-FABP in human coronary artery endothelial cells increased inflammatory cytokines and reduced phosphorylation of nitric oxide synthase 3 4. Ectopic A-FABP increased proliferation and migration of human coronary smooth muscle cells and vascular endothelial dysfunction, which were attenuated by treatment with anti-A-FABP antibody	Fuseya T et al. [56]
2018	Carotid atherosclerosis	281 subjects without medication followed-up for 3 years	1. Serum A-FABP levels were significantly correlated with CIMT 2. Yearly changes in CIMT were positively associated with baseline levels of A-FABP	Furuhashi M et al. [57]
2017	Acute ischemic stroke	737 patients with acute ischemic stroke	1. A-FABP levels were associated with poor functional outcome and mortality 2. Addition of A-FABP improved the prognostic accuracy of National Institutes of Health Stroke Scale score	Tu WJ et al. [58]
2020	Ischemic stroke	30 patients with acute ischemic stroke; A-FABP knockout or wild-type mice subjected to middle cerebral artery occlusion;	1. A-FABP levels were correlated with cerebral infarct volume and levels of matrix metalloproteinases-9 in patients with ischemic stroke 2. Ischemia-induced elevation of A-FABP in macrophages and microglial cells contributed to degradation of tight junction proteins and blood-brain barrier leakage by inducing metalloproteinases-9 expression	Liao B et al. [23]
2020	Heart failure; atherosclerotic cardiovascular diseases	176 patients with type 2 diabetes without established CVD followed-up for 28 months	1. A-FABP levels at baseline was associated with the development of left ventricular hypertrophy and diastolic dysfunction 2. A-FABP levels at baseline predicted the development of major adverse cardiovascular events (composite of cardiovascular death, hospitalization for heart failure, non-fatal myocardial infarction, and stroke)	Wu MZ et al. [59]
2020	Metabolic syndrome; coronary artery disease	37 metabolic syndrome patients undergoing coronary artery bypass grafting (CABG) for underlying coronary artery disease and 23 patients without CAD undergoing heart valve surgery (control group)	1. A-FABP mRNA expression in epicardial adipose tissue was significantly elevated in patients with metabolic syndrome and coronary artery disease 2. The extent of coronary atherosclerosis was significantly associated with the level of expression of A-FABP mRNA in epicardial adipose tissue	Gormez et al. [60]
2021	Heart failure	50 patients with heart failure with preserved ejection fraction and 150 patients with elevated cardiometabolic risk	1. A-FABP levels were associated with important hemodynamic indices, including higher central systolic and diastolic blood pressures 2. Compared to central hemodynamic information alone, the addition of A-FABP levels improved HF risk classification	Yen et al. [61]

Abbreviations used in Table 1: A-FABP, adipocyte-fatty acid binding protein; HbA1c, glycated hemoglobin 1c; IL-6, interleukin-6; ER, endoplasmic reticulum; HFD, high-fat diet; CVD, cardiovascular disease; CIMT, carotid intima-media thickness.

**Table 2 ijms-22-09386-t002:** Summary of human studies showing an association between adipocyte-fatty acid binding protein (A-FABP) and atherosclerotic cardiovascular disease.

Published Year	Cohort	Country	Follow-Up Years	Main Findings	Conclusion	Reference
2007	479 subjects	China	/	1. Serum A-FABP levels were higher in women than in men 2. Serum A-FABP levels were positively correlated with CIMT in both sexes, but an independent association was only observed in women 3. Serum A-FABP levels were independently associated with age and hypertension in women	A-FABP levels are independently associated with carotid atherosclerosis in women	Yeung DC et al. [110]
2010	125 subjects with CAD and 120 control subjects	Japan	/	1. CAD patients had higher A-FABP levels compared to controls 2. Serum A-FABP levels were independently associated with plaque volume in CAD patients 3. Serum A-FABP levels were positively correlated with BMI, IL-6, and hsCRP, and were negatively correlated with HDL-cholesterol and serum adiponectin in CAD patients	Increased serum A-FABP is significantly associated with a greater coronary plaque burden	Miyoshi T et al. [20]
2013	1847 subjects without previous CVD	China	12 years	1. Higher baseline levels of A-FABP were associated with development of CVD 2. Addition of A-FABP to the traditional risk factor model improved the predictive performance	Circulating A-FABP level independently predicts the development of CVD	Chow WS et al. [79]
2013	104 overweight/obese women (BMI ≥ 25 kg/m2) and 76 age-matched healthy controls (BMI < 25 kg/m2)	Poland	/	1. A-FABP concentration was correlated with insulin resistance 2. A-FABP was an independent predictor of triglyceride and HDL-cholesterol 3. A-FABP discriminated overweight/obese patients from healthy individuals	A-FABP is a predictor of atherogenic risk profile	Mankowska-Cyl A et al. [82]
2014	2253 CVD-free subjects with normal glucose tolerance	China	/	A-FABP levels correlated with CIMT in men and in women (both premenopausal and postmenopausal), but an independent association was only observed in women	Serum A-FABP levels are independently associated with subclinical atherosclerosis in pre- and post-menopausal women with normal glucose tolerance	Hao Y et al. [111]
2018	170 subjects with newly diagnosed type 2 diabetes	China	8 years	Patients with higher baseline levels of A-FABP had an increased risk of developing subclinical atherosclerosis at 8 years	Circulating A-FABP levels independently predict the development of subclinical atherosclerosis in type 2 diabetes patients	Xiao Y et al. [114]
2018	281 subjects without medication	Japan	3 years	1. Serum A-FABP levels were significantly correlated with CIMT 2. Yearly changes in CIMT were positively associated with baseline levels of A-FABP	A-FABP concentration is an independent predictor of the progression of carotid atherosclerosis	Furuhashi M et al. [57]
2020	176 patients with type 2 diabetes without established CVD followed-up for 28 months	China	28 months	1. A-FABP levels at baseline was associated with the development of left ventricular hypertrophy and diastolic dysfunction 2. A-FABP levels at baseline predicted the development of major adverse cardiovascular events (composite of cardiovascular death, hospitalization for heart failure, non-fatal myocardial infarction, and stroke)	A-FABP is able to predict adverse cardiovascular outcomes in diabetic patients	Wu MZ et al. [59]

Abbreviations used in Table 2: A-FABP, adipocyte-fatty acid binding protein; CIMT, carotid intima-media thickness; CAD, coronary artery disease; BMI, body mass index; IL-6, Interleukin-6; hsCRP, high-sensitive C-reactive protein; HDL, high-density lipoprotein; CVD, cardiovascular disease.

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
