# Peer review of "A-FABP in Metabolic Diseases and the Therapeutic Implications: An Update"

_ijms, 2021, doi:10.3390/ijms22179386_

Round 1

Reviewer 1 Report

Well prepared, comprehensive review concerning on the role of A-FABP in metabolic diseases. I propose to add some more actual information and  references(2020-21) concerning on the role of A-FABP in cardiovascular diseases, obesity, liver failure, cancerogenesis.

Reviewer 2 Report

Dear authors,

This was a clear and comprehensive review article about cardio-metabolic relationship between A- FABP level and clinical events and therapeutic application.

Three small questions were raised.

  1. The mechanism in hypertensive populations had multi-factorial effect and compiling renin-angiotensin-axis system, sympathetic tone, vasoconstriction and volume overloading. The authors can make more clear and detail  narration.
  2. Atherosclerosis was complicated  mechanism including metabolic risk such as DM, dyslipidemia and inflammation. However, smoking effect was also important.
  3. The table mentioned about previous import reviewed studies were lacking some information, such as case numbers. 
